# Unsupervised Meta-Learning via Few-shot Pseudo-supervised Contrastive Learning

**Huiwon Jang**[A][*]  **Hankook Lee**[B][*][†]  **Jinwoo Shin**[A]
[A]Korea Advanced Institute of Science and Technology (KAIST)    [B]LG AI Research
{huiwoen0516, jinwoos}@kaist.ac.kr  hankook.lee@lgresearch.ai

## Abstract

*Unsupervised meta-learning* aims to learn generalizable knowledge across a distribution of tasks constructed from unlabeled data. Here, the main challenge is how to construct diverse tasks for meta-learning without label information; recent works have proposed to create, e.g., pseudo-labeling via pretrained representations or creating synthetic samples via generative models. However, such a task construction strategy is fundamentally limited due to heavy reliance on the immutable pseudo-labels during meta-learning and the quality of the representations or the generated samples. To overcome the limitations, we propose a simple yet effective unsupervised meta-learning framework, coined *Pseudo-supervised Contrast (PsCo)*, for few-shot classification. We are inspired by the recent self-supervised learning literature; PsCo utilizes a *momentum network* and a *queue* of previous batches to improve pseudo-labeling and construct diverse tasks in a progressive manner. Our extensive experiments demonstrate that PsCo outperforms existing unsupervised meta-learning methods under various in-domain and cross-domain few-shot classification benchmarks. We also validate that PsCo is easily scalable to a large-scale benchmark, while recent prior-art meta-schemes are not.

## 1 Introduction

*Learning to learn* (Thrun & Pratt, 1998), also known as *meta-learning*, aims to learn general knowledge about how to solve unseen, yet relevant tasks from prior experiences solving diverse tasks. In recent years, the concept of meta-learning has found various applications, e.g., few-shot classification (Snell et al., 2017; Finn et al., 2017), reinforcement learning (Duan et al., 2017; Houthooft et al., 2018; Alet et al., 2020), hyperparameter optimization (Franceschi et al., 2018), and so on. Among them, *few-shot classification* is arguably the most popular one, whose goal is to learn some knowledge to classify test samples of unseen classes during (meta-)training with few labeled samples. The common approach is to construct a distribution of few-shot classification (i.e., $N$-way $K$-shot) tasks and optimize a model to generalize across tasks (sampled from the distribution) so that it can rapidly adapt to new tasks. This approach has shown remarkable performance in various few-shot classification tasks but suffers from limited scalability as the task construction phase typically requires a large number of human-annotated labels.

To mitigate the issue, there have been several recent attempts to apply meta-learning to *unlabeled data*, i.e., *unsupervised meta-learning (UML)* (Hsu et al., 2019; Khodadadeh et al., 2019; 2021; Lee et al., 2021; Kong et al., 2021). To perform meta-learning without labels, the authors have suggested various ways to construct synthetic tasks. For example, pioneering works (Hsu et al., 2019; Khodadadeh et al., 2019) assigned pseudo-labels via data augmentations or clustering based on pretrained representations. In contrast, recent approaches (Khodadadeh et al., 2021; Lee et al., 2021; Kong et al., 2021) utilized generative models to generate synthetic (in-class) samples or learn unknown labels via categorical latent variables. They have achieved moderate performance in few-shot learning benchmarks, but are fundamentally limited as: (a) the pseudo-labeling strategies are fixed during meta-learning and impossible to correct mislabeled samples; (b) the generative approaches heavily rely on the quality of generated samples and are cumbersome to scale into large-scale setups.

---

[*]Equal contributions
[†]Work done at KAIST

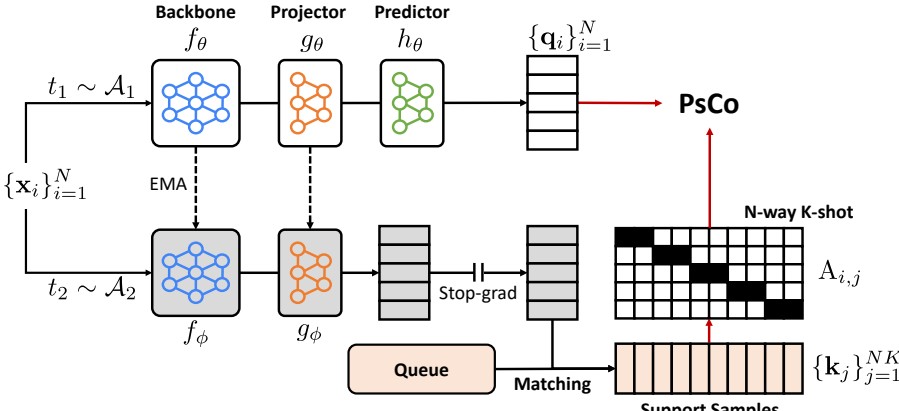

Figure 1: An overview of the proposed **P**seudo-**s**upervised **Co**ntrast (PsCo). PsCo constructs an $N$-way $K$-shot few-shot classification task using the current mini-batch $\{\mathbf{x}_i\}$ and the queue of previous mini-batches; and then, it learns the task via contrastive learning. Here, $\mathbf{A}$ is a label assignment matrix found by the Sinkhorn-Knopp algorithm (Cuturi, 2013), $\mathcal{A}$ is a pre-defined augmentation distribution, $f$ is a backbone feature extractor, $g$ and $h$ are projection and prediction MLPs, respectively, and $\phi$ is an exponential moving average (EMA) of the model parameter $\theta$.

To overcome the limitations of the existing UML approaches, in this paper, we ask whether one can (a) progressively improve a pseudo-labeling strategy during meta-learning, and (b) construct more diverse tasks without generative models. We draw inspiration from recent advances in *self-supervised learning* literature (He et al., 2020; Khosla et al., 2020), which has shown remarkable success in representation learning without labeled data. In particular, we utilize (a) a *momentum network* to improve pseudo-labeling progressively via temporal ensemble; and (b) a *momentum queue* to construct diverse tasks using previous mini-batches in an online manner.

Formally, we propose **P**seudo-**s**upervised **Co**ntrast (PsCo), a novel and effective unsupervised meta-learning framework, for few-shot classification. Our key idea is to construct few-shot classification tasks using the current and previous mini-batches based on the momentum network and the momentum queue. Specifically, given a random mini-batch of $N$ unlabeled samples, we treat them as $N$ queries (i.e., test samples) of different $N$ labels, and then select $K$ shots (i.e., training samples) for each label from the queue of previous mini-batches based on representations extracted by the momentum network. To further improve the selection procedure, we utilize top-$K$ sampling after applying a matching algorithm, Sinkhorn-Knopp (Cuturi, 2013). Finally, we optimize our model via supervised contrastive learning (Khosla et al., 2020) for solving the $N$-way $K$-shot task. Remark that our few-shot task construction relies on not only the current mini-batch but also the momentum network and the queue of previous mini-batches. Therefore, our task construction (i.e., pseudo-labeling) strategy (a) is progressively improved during meta-learning with the momentum network, and (b) constructs diverse tasks since the shots can be selected from the entire dataset. Our framework is illustrated in Figure 1.

Throughout extensive experiments, we demonstrate the effectiveness of the proposed framework, PsCo, under various few-shot classification benchmarks. First, PsCo achieves state-of-the-art performance under both Omniglot (Lake et al., 2011) and miniImageNet (Ravi & Larochelle, 2017) few-shot benchmarks; its performance is even competitive with supervised meta-learning methods. Next, PsCo also shows superiority under cross-domain few-shot learning scenarios. Finally, we demonstrate that PsCo is scalable to a large-scale benchmark, ImageNet (Deng et al., 2009).

We summarize our contributions as follows:

- We propose PsCo, an effective unsupervised meta-learning (UML) framework for few-shot classification, which constructs diverse few-shot pseudo-tasks without labels utilizing the momentum network and the queue of previous batches in a progressive manner.
- We achieve state-of-the-art performance on few-shot classification benchmarks, Omniglot (Lake et al., 2011) and miniImageNet (Ravi & Larochelle, 2017). For example, PsCo outperforms the prior art of UML, Meta-SVEBM (Kong et al., 2021), by 5% accuracy gain (58.03→63.26), for 5-way 5-shot tasks of miniImageNet (see Table 1).

- We show that PsCo achieves comparable performance with supervised meta-learning methods in various few-shot classification benchmarks. For example, PsCo achieves 44.01% accuracy for 5-way 5-shot tasks of an unseen domain, Cars (Krause et al., 2013), while supervised MAML (Finn et al., 2017) does 41.17% (see Table 2).
- We validate PsCo is also applicable to a large-scale dataset: e.g., we improve PsCo by 5.78% accuracy gain (47.67→53.45) for 5-way 5-shot tasks of Cars using large-scale unlabeled data, ImageNet (Deng et al., 2009) (see Table 3).

## 2 PRELIMINARIES

### 2.1 PROBLEM STATEMENT: UNSUPERVISED FEW-SHOT LEARNING

The problem of interest in this paper is *unsupervised few-shot learning*, one of the popular unsupervised meta-learning applications. This aims to learn generalizable knowledge without human annotations for quickly adapting to unseen but relevant few-shot tasks. Following the meta-learning literature, we refer to the learning phase as meta-training and the adaptation phase as meta-test.

Formally, we are only able to utilize an unlabeled dataset $\mathcal{D}_{\texttt{meta\_train}} := \{\mathbf{x}_i\}$ during meta-training our model. At the meta-test phase, we transfer the model to new few-shot tasks $\{\mathcal{T}_i\} \sim \mathcal{D}_{\texttt{meta\_test}}$ where each task $\mathcal{T}_i$ aims to classify query samples $\{\mathbf{x}_q\}$ among $N$ labels using support (i.e., training) samples $\mathcal{S} = \{(\mathbf{x}_s, \mathbf{y}_s)\}_{s=1}^{NK}$. We here assume the task $\mathcal{T}_i$ consists of $K$ support samples for each label $\mathbf{y} \in \{1, \dots, N\}$, which is referred to as $N$-way $K$-shot classification. Note that $\mathcal{D}_{\texttt{meta\_train}}$ and $\mathcal{D}_{\texttt{meta\_test}}$ can come from the same domain (i.e., the standard in-domain setting) or different domains (i.e., cross-domain) as suggested by Chen et al. (2019).

### 2.2 CONTRASTIVE LEARNING

Contrastive learning (Oord et al., 2018; Chen et al., 2020a; He et al., 2020; Khosla et al., 2020) aims to learn meaningful representations by maximizing the similarity between similar (i.e., positive) samples, and minimizing the similarity between dissimilar (i.e., negative) samples on the representation space. We first describe a general form of contrastive learning objectives based on the temperature-normalized cross entropy (Chen et al., 2020a; He et al., 2020) and its variant for multiple positives (Khosla et al., 2020) as follows:

$$\mathcal{L}_{\texttt{Contrast}}(\{\mathbf{q}_i\}_{i=1}^N, \{\mathbf{k}_j\}_{j=1}^M, \mathbf{A}; \tau) := -\frac{1}{N} \sum_{i=1}^N \frac{1}{\sum_j \mathbf{A}_{i,j}} \sum_{j=1}^M \mathbf{A}_{i,j} \log \frac{\exp(\mathbf{q}_i^\top \mathbf{k}_j / \tau)}{\sum_{k=1}^M \exp(\mathbf{q}_i^\top \mathbf{k}_k / \tau)}, \quad (1)$$

where $\{\mathbf{q}_i\}$ and $\{\mathbf{k}_j\}$ are $\ell_2$-normalized query and key representations, respectively, $\mathbf{A} \in \{0, 1\}^{NM}$ represents whether $\mathbf{q}_i$ and $\mathbf{k}_j$ are positive ($\mathbf{A}_{i,j} = 1$) or negative ($\mathbf{A}_{i,j} = 0$), and $\tau$ is a hyperparameter for temperature scaling.

Based on the recent observations in the self-supervised learning literature, we also describe a general scheme to construct the query and key representations using data augmentations and a momentum network. Formally, given a random mini-batch $\{\mathbf{x}_i\}$, the representations can be obtained as follows:

$$\mathbf{q}_i = \text{Normalize}(h_\theta \circ g_\theta \circ f_\theta(t_{i,1}(\mathbf{x}_i))), \qquad \mathbf{k}_i = \text{Normalize}(g_\phi \circ f_\phi(t_{i,2}(\mathbf{x}_i))), \qquad (2)$$

where Normalize($\cdot$) is $\ell_2$ normalization, $t_{i,1} \sim \mathcal{A}_1$ and $t_{i,2} \sim \mathcal{A}_2$ are random data augmentations, $f$ is a backbone feature extractor like ResNet (He et al., 2016), $g$ and $h$ are projection and prediction MLPs,[1] respectively, and $\phi$ is an exponential moving average (i.e., momentum) of the model parameter $\theta$.[2] Since a large number of negative samples plays a crucial role in contrastive learning, one can re-use the key representations of previous mini-batches by maintaining a queue (He et al., 2020).

Note that the above forms (1) and (2) can be formulated as various contrastive learning frameworks. For example, SimCLR (Chen et al., 2020a) is a special case of no momentum $\phi$ and no predictor $h$. In addition, self-supervised contrastive learning methods (Chen et al., 2020a; He et al., 2020) often assume that $\mathbf{k}_i$ is only the positive key of $\mathbf{q}_i$, i.e., $\mathbf{A}_{i,j} = 1$ if and only if $i = j$, while supervised contrastive learning (Khosla et al., 2020) directly uses labels for $\mathbf{A}$.

---

[1]The prediction MLPs have been utilized in the recent SSL literature (Grill et al., 2020; Chen et al., 2021).

[2]$\phi$ is updated by $\phi \leftarrow m\phi + (1 - m)\theta$ for each training iteration where $m$ is a momentum hyperparameter.

---

**Algorithm 1 P**seudo-**s**upervised **Co**ntrast (PsCo): PyTorch-like Pseudocode

---

```
# f, g, h: backbone, projector, and predictor
# {f,g}_ema: momentum backbone, and projector
# queue: momentum queue (Mxd)
# mm: matrix multiplication, mul: element-wise multiplication

def PsCo(x):                            # x: a mini-batch of N samples
    x1, x2 = aug1(x), aug2(x)           # two augmented views of x
    q = h(g(f(x1)))                     # (Nxd) N query representations
    z = g_ema(f_ema(x2))                # (Nxd) N query momentum representations
    sim = mm(z, queue.T)                # (NxM) similarity matrix
    A_tilde = sinkhorn(sim)             # (NxM) soft pseudo-label assignment matrix
    s, A = select_topK(queue, A_tilde)  # (NKxd) s: support momentum representations
                                        # (NxNK) A: pseudo-label assignment matrix
    logits = mm(q, s.T) / temperature
    loss = logits.logsumexp(dim=1) - mul(logits, A).sum(dim=1) / K
    return loss.mean()
```

---

## 3    METHOD: PSEUDO-SUPERVISED CONTRASTIVE META-LEARNING

In this section, we introduce ***Pseudo-supervised Contrast*** (*PsCo*), a novel and effective framework for unsupervised few-shot learning. Our key idea is to construct few-shot classification pseudo-tasks using the current and previous mini-batches with the momentum network and the momentum queue. We then employ supervised contrastive learning (Khosla et al., 2020) for learning the pseudo-tasks. The detailed implementations of our task construction, meta-training objective, and meta-test scheme for unsupervised few-shot learning are described in Section 3.1, 3.2, and 3.3, respectively. Our framework is illustrated in Figure 1 and its pseudo-code is provided in Algorithm 1. Note that we use the same notations described in Section 2 for consistency.

### 3.1    ONLINE PSEUDO-TASK CONSTRUCTION

We here describe how to construct a few-shot pseudo-task using unlabeled data $\mathcal{D}_{\texttt{meta\_train}} = \{\mathbf{x}_i\}$. To this end, we maintain a queue of previous mini-batches. Then, we treat the previous and current mini-batch samples as training (i.e., shots) and test (i.e., queries) samples for our few-shot pseudo-task. Formally, let $\mathcal{B} := \{\mathbf{x}_i\}_{i=1}^{N}$ be the current mini-batch randomly sampled from $\mathcal{D}_{\texttt{meta\_train}}$, and $\mathcal{Q} := \{\tilde{\mathbf{x}}_j\}_{j=1}^{M}$ be the queue of previous mini-batch samples. Now, we treat $\mathcal{B} = \{\mathbf{x}_i\}_{i=1}^{N}$ as queries of $N$ different pseudo-labels and find $K$ (appropriate) shots for each pseudo-label from the queue $\mathcal{Q}$. Remark that this approach to utilize the previous mini-batches encourages us to construct more diverse tasks.

To find the shots efficiently, we utilize the momentum network and the momentum queue described in Section 2.2. For the current mini-batch samples, we compute the momentum query representations with data augmentations $t_{i,2} \sim \mathcal{A}_2$, i.e., $\mathbf{z}_i := \text{Normalize}(g_\phi \circ f_\phi(t_{i,2}(\mathbf{x}_i)))$. Following He et al. (2020), we store only the momentum representations of the previous mini-batch samples instead of raw data in the queue $\mathcal{Q}_{\mathbf{z}}$, i.e., $\mathcal{Q}_{\mathbf{z}} := \{\tilde{\mathbf{z}}_j\}_{j=1}^{M}$. Remark that the use of the momentum network is not only for efficiency but also for improving our task construction strategy because the momentum network is consistent and progressively improved during training. Following He et al. (2020), we randomly initialize the queue $\mathcal{Q}_{\mathbf{z}}$ at the beginning of training.

Now, the remaining question is as follows: How to find $K$ appropriate shots from the queue $\mathcal{Q}$ for each pseudo-label using the momentum representations? Before introducing our algorithm, we first discuss two requirements for constructing semantically meaningful few-shot tasks: (i) shots and queries of the same label should be semantically similar, and (ii) all shots should be different. Based on these requirements, we formulate our assignment problem as follows:

$$\max_{\tilde{\mathbf{A}} \in \{0,1\}^{N \times M}} \sum_{i=1}^{N} \sum_{j=1}^{M} \tilde{\mathrm{A}}_{ij} \cdot \mathbf{z}_i^\top \tilde{\mathbf{z}}_j \quad \text{such that} \quad \sum_j \tilde{\mathrm{A}}_{ij} = K, \quad \sum_i \tilde{\mathrm{A}}_{ij} \leq 1. \tag{3}$$

Obtaining the exact optimal solution to the above assignment problem for each training iteration might be too expensive for our purpose (Ramshaw & Tarjan, 2012). Instead, we use an approximate algorithm: we first apply a fast version (Cuturi, 2013) of the Sinkhorn-Knopp algorithm to solve the

following problem:

$$\max_{\tilde{\mathbf{A}} \in [0,1]^{N \times M}} \sum_{i=1}^{N} \sum_{j=1}^{M} \tilde{\mathbf{A}}_{ij} \cdot \mathbf{z}_i^\top \tilde{\mathbf{z}}_j + \epsilon H(\tilde{\mathbf{A}}) \quad \text{such that} \quad \sum_j \tilde{\mathbf{A}}_{ij} = 1/N, \quad \sum_i \tilde{\mathbf{A}}_{ij} = 1/M, \quad (4)$$

which is an entropy-regularized optimal transport problem (Cuturi, 2013). Its optimal solution $\tilde{\mathbf{A}}^*$ can be obtained efficiently and can be considered as a soft assignment matrix between the current mini-batch $\{\mathbf{z}_i\}_{i=1}^N$ and the queue $\mathcal{Q}_{\mathbf{z}} = \{\tilde{\mathbf{z}}_j\}_{j=1}^M$. Hence, we select top-$K$ elements for each row of the assignment matrix $\tilde{\mathbf{A}}^*$ and finally construct an $N$-way $K$-shot pseudo-task consisting of (a) query samples $\mathcal{B} = \{\mathbf{x}_i\}_{i=1}^N$, (b) the support representations $\mathcal{S}_{\mathbf{z}} := \{\tilde{\mathbf{z}}_s\}_{s=1}^{NK}$, and (c) the pseudo-label assignment matrix $\mathbf{A} \in \{0,1\}^{N \times NK}$. Note that Figure 1 shows an example of a 5-way 2-shot task. We empirically observe that our task construction strategy satisfies the above requirements (i) and (ii) (see Section 4.3).

## 3.2 META-TRAINING: SUPERVISED CONTRASTIVE LEARNING WITH PSEUDO TASKS

We now describe our meta-learning objective $\mathcal{L}_{\texttt{PsCo}}$ for learning our few-shot pseudo-tasks. We here use our model $\theta$ to obtain query representations: $\mathbf{q}_i := \text{Normalize}(h_\theta \circ g_\theta \circ f_\theta(t_{i,1}(\mathbf{x}_i)))$ where $t_{i,1} \sim \mathcal{A}_1$ is a random data augmentation for each $i$. Then, our objective $\mathcal{L}_{\texttt{PsCo}}$ is defined as follows:

$$\mathcal{L}_{\texttt{PsCo}} := \mathcal{L}_{\texttt{Contrast}}(\{\mathbf{q}_i\}_{i=1}^N, \mathcal{S}_{\mathbf{z}}, \mathbf{A}; \tau_{\texttt{PsCo}}), \quad (5)$$

where $\mathcal{S}_{\mathbf{z}} := \{\tilde{\mathbf{z}}_s\}_{s=1}^{NK}$ is the support representations and $\mathbf{A} \in \{0,1\}^{N \times NK}$ is the pseudo-label assignment matrix, which are constructed by our task construction strategy described in Section 3.1.

Since our framework PsCo uses the same architectural components as a self-supervised learning framework, MoCo (He et al., 2020), the MoCo objective $\mathcal{L}_{\texttt{MoCo}}$ can be incorporated into our PsCo without additional computation costs. Note that the MoCo objective can be written as follows:

$$\mathcal{L}_{\texttt{MoCo}} := \mathcal{L}_{\texttt{Contrast}}(\{\mathbf{q}_i\}_{i=1}^N, \{\mathbf{z}_i\}_{i=1}^N \cup \mathcal{Q}_{\mathbf{z}}, \mathbf{A}_{\texttt{MoCo}}; \tau_{\texttt{MoCo}}), \quad (6)$$

where $(\mathbf{A}_{\texttt{MoCo}})_{i,j} = 1$ if and only if $i = j$, and $\mathbf{z}_i := \text{Normalize}(g_\phi \circ f_\phi(t_{i,2}(\mathbf{x}_i)))$ as described in Section 3.1. We optimize our model $\theta$ via all the objectives, i.e., $\mathcal{L}_{\texttt{total}} := \mathcal{L}_{\texttt{PsCo}} + \mathcal{L}_{\texttt{MoCo}}$. Remark again that $\phi$ is updated by exponential moving average (EMA), i.e., $\phi \leftarrow m\phi + (1-m)\theta$.

**Weak augmentation for momentum representations.** To successfully find the pseudo-label assignment matrix $\mathbf{A}$, we apply weak augmentations for the momentum representations (i.e., $\mathcal{A}_2$ is weaker than $\mathcal{A}_1$) as Zheng et al. (2021) did. This reduces the noise in the representations and consequently enhances the performance of our PsCo as $\mathbf{A}$ becomes more accurate (see Section 4.3).

## 3.3 META-TEST

At the meta-test stage, we have an $N$-way $K$-shot task $\mathcal{T}$ consisting of query samples $\{\mathbf{x}_q\}$ and support samples $\mathcal{S} = \{(\mathbf{x}_s, \mathbf{y}_s)\}_{s=1}^{NK}$.[3] We here discard the momentum network $\phi$ and use only the online network $\theta$. To predict labels, we first compute the query representation $\mathbf{q}_q := \text{Normalize}(h_\theta \circ g_\theta \circ f_\theta(\mathbf{x}_q))$ and the support representations $\mathbf{z}_s := \text{Normalize}(g_\theta \circ f_\theta(\mathbf{x}_s))$. Then we predict a label by the following classification rule: $\hat{y} := \arg\max_y \mathbf{q}_q^\top \mathbf{c}_y$ where $\mathbf{c}_y := \text{Normalize}(\sum_s \mathbf{1}_{y_s = y} \cdot \mathbf{z}_s)$ is the prototype vector. This is inspired by our $\mathcal{L}_{\texttt{PsCo}}$, which can be interpreted as minimizing distance from the mean (i.e., prototype) of the shot representations.[4]

**Further adaptation for cross-domain few-shot classification.** Under cross-domain few-shot classification scenarios, the model $\theta$ should further adapt to the meta-test domain due to the dissimilarity from meta-training. We here suggest an efficient adaptation scheme using only a few labeled samples. Our idea is to consider the support samples as queries. To be specific, we compute the query representation $\mathbf{q}_s := \text{Normalize}(h_\theta \circ g_\theta \circ f_\theta(\mathbf{x}_s))$ for each support sample $\mathbf{x}_s$, and construct the label assignment matrix $\mathbf{A}'$ as $\mathbf{A}'_{s,s'} = 1$ if and only if $y_s = y_{s'}$. Then we simply optimize only $g_\theta$ and $h_\theta$ via contrastive learning, i.e., $\mathcal{L}_{\texttt{Contrast}}(\{\mathbf{q}_s\}, \{\mathbf{z}_s\}, \mathbf{A}'; \tau_{\texttt{PsCo}})$, for few iterations. We empirically observe that this adaptation scheme is effective under cross-domain settings (see Section 4.3).

---

[3]Note that $N$ and $K$ for meta-training and meta-test could be different. We use a large $N$ (e.g., $N = 256$) during meta-training to fully utilize computational resources like standard deep learning, and a small $N$ (e.g., $N = 5$) during meta-test following the meta-learning literature.

[4]$\mathcal{L}_{\texttt{PsCo}} = -\frac{1}{N} \sum_i \frac{1}{\tau_{\texttt{PsCo}}} \mathbf{q}_i^\top \left( \frac{1}{K} \sum_j \mathbf{A}_{i,j} \mathbf{z}_j \right) + \text{term not depending on } \mathbf{A}.$

Table 1: Few-shot classification accuracy (%) on Omniglot and miniImageNet benchmarks. We report the average accuracy over 2000 few-shot tasks for PsCo and self-supervised learning methods. Other reported numbers borrow from Khodadadeh et al. (2021); Kong et al. (2021). **Bold** entries indicate the best for each task configuration, among unsupervised and self-supervised methods.

| Method | Omniglot (way, shot) | | | | miniImageNet (way, shot) | | | |
|---|---|---|---|---|---|---|---|---|
| | (5,1) | (5,5) | (20,1) | (20,5) | (5,1) | (5,5) | (5,20) | (5,50) |
| *Training from Scratch* | 52.50 | 74.78 | 24.91 | 47.62 | 27.59 | 38.48 | 51.53 | 59.63 |
| *Unsupervised meta-learning* | | | | | | | | |
| CACTUs-MAML | 68.84 | 87.78 | 48.09 | 73.36 | 39.90 | 53.97 | 63.84 | 69.64 |
| CACTUs-ProtoNets | 68.12 | 83.58 | 47.75 | 66.27 | 39.18 | 53.36 | 61.54 | 63.55 |
| UMTRA | 83.80 | 95.43 | 74.25 | 92.12 | 39.93 | 50.73 | 61.11 | 67.15 |
| LASIUM-MAML | 83.26 | 95.29 | - | - | 40.19 | 54.56 | 65.17 | 69.13 |
| LASIUM-ProtoNets | 80.15 | 91.10 | - | - | 40.05 | 52.53 | 61.09 | 64.89 |
| Meta-GMVAE | 94.92 | 97.09 | 82.21 | 90.61 | 42.82 | 55.73 | 63.14 | 68.26 |
| Meta-SVEBM | 91.85 | 97.21 | 79.66 | 92.21 | 43.38 | 58.03 | 67.07 | 72.28 |
| **PsCo (Ours)** | **96.37** | **99.13** | **89.64** | **97.07** | **46.70** | **63.26** | **72.22** | **73.50** |
| *Self-supervised learning* | | | | | | | | |
| SimCLR | 92.13 | 97.06 | 80.95 | 91.60 | 43.35 | 52.50 | 61.83 | 64.85 |
| MoCo v2 | 92.66 | 97.38 | 82.13 | 92.35 | 41.92 | 50.94 | 60.23 | 63.45 |
| SwAV | 93.13 | 97.32 | 82.63 | 92.12 | 43.24 | 52.41 | 61.36 | 64.52 |
| *Supervised meta-learning* | | | | | | | | |
| MAML | 94.46 | 98.83 | 84.60 | 96.29 | 46.81 | 62.13 | 71.03 | 75.54 |
| ProtoNets | 98.35 | 99.58 | 95.31 | 98.81 | 46.56 | 62.29 | 70.05 | 72.04 |

## 4 EXPERIMENTS

In this section, we demonstrate the effectiveness of the proposed framework under standard few-shot learning benchmarks (Section 4.1) and cross-domain few-shot learning benchmarks (Section 4.2). We provide ablation studies regarding PsCo in Section 4.3. Following Lee et al. (2021), we mainly use Conv4 and Conv5 architectures for Omniglot (Lake et al., 2011) and miniImageNet (Ravi & Larochelle, 2017), respectively, for the backbone feature extractor $f_\theta$. For the number of shots during meta-learning, we use $K = 1$ for Omniglot and $K = 4$ for miniImageNet (see Table 6 for the sensitivity of $K$). Other details are fully described in Appendix A. We omit the confidence intervals in this section for clarity, and the full results with them are provided in Appendix F.

### 4.1 STANDARD FEW-SHOT BENCHMARKS

**Setup.** We here evaluate PsCo on the standard few-shot benchmarks of unsupervised meta-learning: Omniglot (Lake et al., 2011) and miniImageNet (Ravi & Larochelle, 2017). We compare PsCo's performance with unsupervised meta-learning methods (Hsu et al., 2019; Khodadadeh et al., 2019; 2021; Lee et al., 2021; Kong et al., 2021), self-supervised learning methods (Chen et al., 2020a;b; Caron et al., 2020), and supervised meta-learning methods (Finn et al., 2017; Snell et al., 2017) on the benchmarks. The details of the benchmarks and the baselines are described in Appendix D.

**Few-shot classification results.** Table 1 shows the results of the few-shot classification with various (way, shot) tasks of Omniglot and miniImageNet. PsCo achieves state-of-the-art performance on both Omniglot and miniImageNet benchmarks under the unsupervised setting. For example, we obtain 5% accuracy gain (67.07 → 72.22) on miniImageNet 5-way 20-shot tasks. Moreover, the performance is even competitive with supervised meta-learning methods, ProtoNets (Snell et al., 2017), and MAML (Finn et al., 2017) as well.

### 4.2 CROSS-DOMAIN FEW-SHOT BENCHMARKS

**Setup.** We evaluate PsCo on cross-domain few-shot classification benchmarks following Oh et al. (2022). To be specific, we use (a) benchmark of large-similarity with ImageNet: CUB (Wah et al., 2011), Cars (Krause et al., 2013), Places (Zhou et al., 2018), and Plantae (Horn et al., 2018); (b) benchmarks of small-similarity with ImageNet: CropDiseases (Mohanty et al., 2016), EuroSAT (Helber et al., 2019), ISIC (Codella et al., 2018), and ChestX (Wang et al., 2017). As baselines, we

Table 2: Few-shot classification accuracy (%) on cross-domain few-shot classification benchmarks. We transfer Conv5 trained on miniImageNet to each benchmark. We report the average accuracy over 2000 few-shot tasks for all methods, except Meta-SVEBM as it is evaluated over 200 tasks due to the long evaluation time. **Bold** entries indicate the best for each task configuration, among unsupervised and self-supervised methods.

(a) Cross-domain few-shot benchmarks similar to miniImageNet.

| | CUB | | Cars | | Places | | Plantae | |
|---|---|---|---|---|---|---|---|---|
| **Method** | **(5, 5)** | **(5, 20)** | **(5, 5)** | **(5, 20)** | **(5, 5)** | **(5, 20)** | **(5, 5)** | **(5, 20)** |
| *Unsupervised meta-learning* | | | | | | | | |
| Meta-GMVAE | 47.48 | 54.08 | 31.39 | 38.36 | 57.70 | 65.08 | 38.27 | 45.02 |
| Meta-SVEBM | 45.50 | 54.61 | 34.27 | 46.23 | 51.27 | 61.09 | 38.12 | 46.22 |
| **PsCo (Ours)** | **57.38** | **68.58** | **44.01** | **57.50** | **63.60** | **73.95** | **52.72** | **64.53** |
| *Self-supervised learning* | | | | | | | | |
| SimCLR | 52.11 | 61.89 | 37.40 | 50.05 | 60.10 | 69.93 | 43.42 | 54.92 |
| MoCo v2 | 53.23 | 62.81 | 38.65 | 51.77 | 59.09 | 69.08 | 43.97 | 55.45 |
| SwAV | 51.58 | 61.38 | 36.85 | 50.03 | 59.57 | 69.70 | 42.68 | 54.03 |
| *Supervised meta-learning* | | | | | | | | |
| MAML | 56.57 | 64.17 | 41.17 | 48.82 | 60.05 | 67.54 | 47.33 | 54.86 |
| ProtoNets | 56.74 | 65.03 | 38.98 | 47.98 | 59.39 | 67.77 | 45.89 | 54.29 |

(b) Cross-domain few-shot benchmarks dissimilar to miniImageNet.

| | CropDiseases | | EuroSAT | | ISIC | | ChestX | |
|---|---|---|---|---|---|---|---|---|
| **Method** | **(5, 5)** | **(5, 20)** | **(5, 5)** | **(5, 20)** | **(5, 5)** | **(5, 20)** | **(5, 5)** | **(5, 20)** |
| *Unsupervised meta-learning* | | | | | | | | |
| Meta-GMVAE | 73.56 | 81.22 | 73.83 | 80.11 | 33.48 | 39.48 | 23.23 | 26.26 |
| Meta-SVEBM | 71.82 | 83.13 | 70.83 | 80.21 | 38.85 | 48.43 | **26.26** | 28.91 |
| **PsCo (Ours)** | **88.24** | **94.95** | **81.08** | **87.65** | **44.00** | **54.59** | 24.78 | 27.69 |
| *Self-supervised learning* | | | | | | | | |
| SimCLR | 79.90 | 88.73 | 79.14 | 85.05 | 42.83 | 51.35 | 25.14 | **29.21** |
| MoCo v2 | 80.96 | 89.85 | 79.94 | 86.16 | 43.43 | 52.14 | 25.24 | 29.19 |
| SwAV | 80.15 | 89.24 | 79.31 | 85.62 | 43.21 | 51.99 | 24.99 | 28.57 |
| *Supervised meta-learning* | | | | | | | | |
| MAML | 77.76 | 83.24 | 71.48 | 76.70 | 47.34 | 55.09 | 22.61 | 24.25 |
| ProtoNets | 76.01 | 83.64 | 64.91 | 70.88 | 40.62 | 48.38 | 23.15 | 25.72 |

test the previous state-of-the-art unsupervised meta-learning (Lee et al., 2021; Kong et al., 2021), self-supervised learning (Chen et al., 2020a;b; Caron et al., 2020), and supervised meta-learning (Finn et al., 2017; Snell et al., 2017). We here use our adaptation scheme (Section 3.3) with 50 iterations. The details of the benchmarks and implementations are described in Appendix E.

**Small-scale cross-domain few-shot classification results.** We here evaluate various Conv5 models meta-trained on miniImageNet as used in Section 4.1. Table 2 shows that PsCo outperforms all the baselines across all the benchmarks, except ChestX, which is too different from the distribution of miniImageNet (Oh et al., 2022). Somewhat interestingly, PsCo competitive with supervised learning under these benchmarks, e.g., PsCo achieves 88% accuracy on CropDiseases 5-way 5-shot tasks, whereas MAML gets 77%. This implies that our unsupervised method, PsCo, generalizes on more diverse tasks than supervised learning, which is specialized to in-domain tasks.

**Large-scale cross-domain few-shot classification results.** We also validate that our meta-learning framework is applicable to the large-scale benchmark, ImageNet (Deng et al., 2009). Remark that the recent unsupervised meta-learning methods (Lee et al., 2021; Kong et al., 2021; Khodadadeh et al., 2021) rely on generative models, so they are not easily applicable to such a large-scale benchmark. For example, we observe that PsCo is 2.7 times faster than the best baseline, Meta-SVEBM (Kong et al., 2021), even though Meta-SVEBM uses low-dimensional representations instead of full images during training. Hence, we compare PsCo with (a) self-supervised methods, MoCo v2 (Chen et al., 2020b) and BYOL (Grill et al., 2020), and (b) the publicly-available supervised learning baseline. We here use the ResNet-50 (He et al., 2016) architecture. The training details are described in Appendix E.4 and we also provide ResNet-18 results in Appendix F.

Table 3: 5-way 5-shot classification accuracy (%) on cross-domain few-shot benchmarks. We transfer ImageNet-trained ResNet-50 models to each benchmark. We report the average accuracy over 600 few-shot tasks.

| Method | CUB | Cars | Places | Plantae | CropDiseases | EuroSAT | ISIC | ChestX |
|---|---|---|---|---|---|---|---|---|
| MoCo v2 | 64.16 | 47.67 | 81.39 | 61.36 | 82.89 | 76.96 | 38.26 | **24.28** |
| +PsCo (Ours) | **76.63** | **53.45** | **83.87** | **69.17** | **89.85** | **83.99** | **41.64** | 23.60 |
| BYOL | 67.45 | 45.74 | 75.43 | 56.86 | 80.82 | 77.70 | 37.27 | 24.15 |
| +PsCo (Ours) | **82.13** | **56.19** | **83.80** | **71.14** | **92.92** | **85.33** | **42.90** | **26.05** |
| Supervised | 89.13 | 75.15 | 84.41 | 72.91 | 90.96 | 85.64 | 43.34 | 25.35 |

Table 4: Component ablation studies on Omniglot.

| Momentum | Predictor | Sinkhorn | Top-K sampling | $\mathcal{L}_{\texttt{MoCo}}$ | (5, 1) | (5, 5) | (20, 1) | (20, 5) |
|---|---|---|---|---|---|---|---|---|
| ✓ | ✓ | ✓ | ✓ | ✓ | **96.37** | **99.13** | **89.64** | **97.07** |
| ✗ | ✓ | ✓ | ✓ | ✓ | 90.32 | 96.78 | 76.17 | 90.41 |
| ✓ | ✗ | ✓ | ✓ | ✓ | 90.21 | 96.86 | 76.15 | 90.53 |
| ✓ | ✓ | ✗ | ✓ | ✓ | 95.81 | 98.94 | 88.25 | 96.57 |
| ✓ | ✓ | ✓ | ✗ | ✓ | 94.95 | 98.81 | 86.32 | 96.05 |
| ✓ | ✓ | ✓ | ✓ | ✗ | 93.16 | 97.40 | 81.03 | 91.33 |

Table 3 shows that (i) PsCo consistently improves both MoCo and BYOL under this setup (e.g., $67\% \rightarrow 82\%$ in CUB), and (ii) PsCo benefits from the large-scale dataset as we obtain a huge amount of performance gain on the benchmarks of large-similarity with ImageNet: CUB, Cars, Places, and Plantae. Consequently, we achieve comparable performance with the supervised learning baseline, except Cars, which shows that our PsCo is applicable to large-scale unlabeled datasets.

## 4.3 ABLATION STUDY

**Component analysis.** In Table 4, we demonstrate the necessity of each component in PsCo by removing the components one by one: momentum encoder $\phi$, prediction head $h$, Sinkhorn-Knopp algorithm, top-$K$ sampling for sampling support samples, and the MoCo objective, $\mathcal{L}_{\texttt{MoCo}}$ (6). We found that the momentum network $\phi$ and the prediction head $h$ are critical architectural components in our framework like recent self-supervised learning frameworks (Grill et al., 2020; Chen et al., 2021). In addition, Table 4 shows that training with only our objective, $\mathcal{L}_{\texttt{PsCo}}$ (5), achieves meaningful performance, but incorporating it into MoCo is more beneficial. To further validate that our task construction is progressively improved during meta-learning, we evaluate whether a query and a corresponding support sample have the same true label. Figure 2a shows that our task construction is progressively improved, i.e., the task requirement (i) described in Section 3.1 satisfies.

Table 4 also verifies the contribution of the Sinkhorn-Knopp algorithm and Top-$K$ sampling for the performance of PsCo. We further analyze the effect of the Sinkhorn-Knopp algorithm by measuring the overlap ratio of selected supports between different pseudo-labels. As shown in Figure 2b, there are almost zero overlaps when using the Sinkhorn-Knopp algorithm, which means the constructed task is a valid few-shot task, satisfying the task requirement (ii) described in Section 3.1.

**Adaptation effect on cross-domain.** To validate the effect of our adaptation scheme (Section 3.3), we evaluate the few-shot classification accuracy during the adaptation process on miniImageNet (i.e., in-domain) and CropDiseases (i.e., cross-domain) benchmarks. As shown in Figure 2d, we found that the adaptation scheme is more useful in cross-domain benchmarks than in-domain ones. Based on these results, we apply the scheme to only the cross-domain scenarios. We also found that our adaptation does not cause over-fitting since we only optimize the projection and prediction heads $g_\theta$ and $h_\theta$. The results for the adaptation effect on the whole benchmarks are represented in Appendix C.

**Augmentations.** We here confirm that weak augmentation for the momentum network (i.e., $\mathcal{A}_2$) is more effective than strong augmentation unlike other self-supervised learning literature (Chen et al., 2020a; He et al., 2020). We denote the standard augmentation consisting of both geometric and color transformations by *Strong*, and a weaker augmentation consisting of only geometric transformations as *Weak* (see details in Appendix A). As shown in Table 5, utilizing the weak augmentation for $\mathcal{A}_2$ is much more beneficial since it helps to find an accurate pseudo-label assignment matrix **A**.

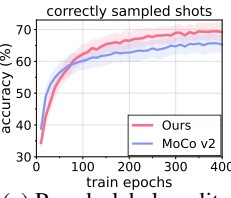 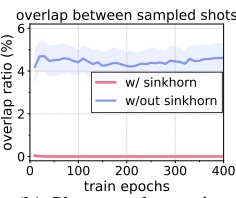 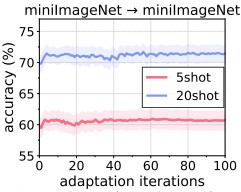 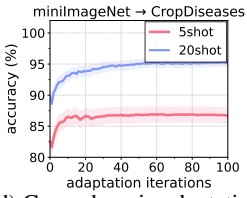

| (a) Pseudo-label quality | (b) Shot overlap ratio | (c) In-domain adaptation | (d) Cross-domain adaptation |
|---|---|---|---|

Figure 2: (a) Pseudo-label quality, measuring the agreement between pseudo-labels and true labels, (b) Shot overlap ratio, measuring whether the shots for each pseudo-label are disjoint, during meta-training. (c,d) Performance while adaptation on in-domain (miniImageNet) and cross-domain (CropDiseases) benchmarks, respectively. We obtain these results from 100 random batches.

Table 5: The ablation study with varying augmentation choices for $\mathcal{A}_1$ and $\mathcal{A}_2$ on miniImageNet.

| $\mathcal{A}_1$ | $\mathcal{A}_2$ | (5, 1) | (5, 5) | (5, 20) | (5, 50) |
|---|---|---|---|---|---|
| Strong | Strong | 44.54 | 60.04 | 68.61 | 71.20 |
| Strong | Weak | **46.70** | **63.26** | **72.22** | **73.50** |
| Weak | Strong | 43.56 | 58.77 | 67.21 | 69.46 |
| Weak | Weak | 40.68 | 55.05 | 63.32 | 65.82 |

Table 6: The ablation study with varying $K$ on miniImageNet.

| $K$ | (5, 1) | (5, 5) | (5, 20) | (5, 50) |
|---|---|---|---|---|
| 1 | 45.88 | 61.84 | 70.25 | 72.76 |
| 4 | **46.70** | **63.26** | **72.22** | **73.50** |
| 16 | 46.31 | 62.76 | 70.91 | 73.43 |
| 64 | 46.60 | 62.50 | 70.82 | 73.22 |

**Training $K$.** We also look at the effect of the training $K$, i.e. number of shots sampled online. We conduct the experiment with $K \in \{1, 4, 16, 64\}$. We observe that PsCo performs consistently well regardless of the choice of $K$ as shown in Table 6. The proper $K$ is suggested to obtain the best-performing models, e.g., $K = 4$ for miniImageNet and $K = 1$ for Omniglot are the best.

## 5 RELATED WORKS

**Unsupervised meta-learning.** Unsupervised meta-learning (Hsu et al., 2019; Khodadadeh et al., 2019; Lee et al., 2021; Kong et al., 2021; Khodadadeh et al., 2021) links meta-learning and unsupervised learning by constructing synthetic tasks and extracting the meaningful information from unlabeled data. For example, CACTUs (Hsu et al., 2019) cluster the data on the pretrained representations at the beginning of meta-learning to assign pseudo-labels. Instead of pseudo-labeling, UMTRA (Khodadadeh et al., 2019) and LASIUM (Khodadadeh et al., 2021) generate synthetic samples using data augmentations or pretrained generative networks like BigBiGAN (Donahue & Simonyan, 2019). Meta-GMVAE (Lee et al., 2021) and Meta-SVEBM (Kong et al., 2021) represent unknown labels via categorical latent variables using variational autoencoders (Kingma & Welling, 2014) and energy-based models (Teh et al., 2003), respectively. In this paper, we suggest a novel online pseudo-labeling strategy to construct diverse tasks without help from any pretrained network or generative model. As a result, our method is easily applicable to large-scale datasets.

**Self-supervised learning.** Self-supervised learning (SSL) (Doersch et al., 2015) has shown remarkable success for unsupervised representation learning across various domains, including vision (He et al., 2020; Chen et al., 2020a), speech (Oord et al., 2018), and reinforcement learning (Laskin et al., 2020). Among SSL objectives, contrastive learning (Oord et al., 2018; Chen et al., 2020a; He et al., 2020) is arguably most popular for learning meaningful representations. In addition, recent advances have been made with the development of various architectural components: e.g., Siamese networks (Doersch et al., 2015), momentum networks (He et al., 2020), and asymmetric architectures (Grill et al., 2020; Chen & He, 2021). In this paper, we utilize the SSL components to construct diverse few-shot tasks in an unsupervised manner.

## 6 CONCLUSION

Although unsupervised meta-learning (UML) and self-supervised learning (SSL) share the same purpose of learning generalizable knowledge to unseen tasks by utilizing unlabeled data, there still exists a gap between UML and SSL literature. In this paper, we bridge the gap as we tailor various SSL components to UML, especially for few-shot classification, and we achieve superior performance under various few-shot classification scenarios. We believe our research could bring many future research directions in both the UML and SSL communities.

## ETHICS STATEMENT

Unsupervised learning, especially self-supervised learning, often requires a large number of training samples, a huge model, and a high computational cost for training the model on large-scale data to obtain meaningful representations because of the absence of human annotations. Furthermore, fine-tuning the model for solving a new task is also time-consuming and memory-inefficient. Hence, it could raise environmental issues such as carbon generation, which could bring an abnormal climate and accelerate global warming. In that sense, meta-learning should be considered as a solution since its purpose is to learn generalizable knowledge that can be quickly adapted to unseen tasks. In particular, unsupervised meta-learning, which benefits from both meta-learning and unsupervised learning, would be an important research direction. We believe that our work could be a useful step toward learning *easily*-generalizable knowledge from unlabeled data.

## REPRODUCIBILITY STATEMENT

We provide all the details to reproduce our experimental results in Appendix A, D, and E. The code is available at https://github.com/alinlab/PsCo. In our experiments, we mainly use NVIDIA GTX3090 GPUs.

## ACKNOWLEDGMENTS AND DISCLOSURE OF FUNDING

This work was mainly supported by Institute of Information & communications Technology Planning & Evaluation (IITP) grant funded by the Korea government (MSIT) (No.2019-0-00075, Artificial Intelligence Graduate School Program (KAIST); No.2022-0-00713, Meta-learning applicable to real-world problems; No.2022-0-00959, Few-shot Learning of Causal Inference in Vision and Language for Decision Making).

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

## A  IMPLEMENTATION DETAILS

We train our models via stochastic gradient descent (SGD) with a batch size of $N = 256$ for 400 epochs. Following Chen et al. (2020b); Chen & He (2021), we use an initial learning rate of 0.03 with the cosine learning schedule, $\tau_{\text{MoCo}} = 0.2$, and a weight decay of $5 \times 10^{-4}$. We use a queue size of $M = 16384$ since Omniglot (Lake et al., 2011) and miniImageNet (Ravi & Larochelle, 2017) has roughly 100k meta-training samples. Following Lee et al. (2021), we use Conv4 and Conv5 for Omniglot and miniImageNet, respectively, for the backbone feature extractor $f_\theta$. We describe the detailed architectures in Table 7. For projection and prediction MLPs, $g_\theta$ and $h_\theta$, we use 2-layer MLPs with a hidden size of 2048 and an output dimension of 128. For the hyperparameters of PsCo, we use $\tau_{\text{PsCo}} = 1$ and a momentum parameter of $m = 0.99$ (see Appendix B for the hyperparameter sensitivity). For the number of shots during meta-learning, we use $K = 1$ for Omniglot and $K = 4$ for miniImageNet (see Table 6 for the sensitivity of $K$). We use the last-epoch model for evaluation without any guidance from the meta-validation dataset.

Table 7: Pytorch-like architecture descriptions for standard few-shot benchmarks

| Backbone | Layer descriptions | Output shape |
|---|---|---|
| Conv4 | `[Conv2d(3×3, 64 filter), BatchNorm2d, ReLU, MaxPool2d(2×2)]×4` | $64 \times 2 \times 2$ |
| Conv5 | `[Conv2d(3×3, 64 filter), BatchNorm2d, ReLU, MaxPool2d(2×2)]×5` | $64 \times 2 \times 2$ |

**Augmentations.** We describe the augmentations for Omniglot and miniImagenet in Table 8. For Omniglot, because it is difficult to apply many augmentations to gray-scale images, we use the same rule for weak and strong augmentations. For miniImageNet, we use only geometric transformations for the weak augmentation following Zheng et al. (2021).

Table 8: Pytorch-like augmentation descriptions for Omniglot and miniImageNet

| Dataset | Augmentation | Descriptions |
|---|---|---|
| Omniglot | Strong, Weak | `RandomResizeCrop(28, scale=(0.2, 1))`
`RandomHorizontalFlip()` |
| miniImagenet | Strong | `RandomResizedCrop(84, scale=(0.2, 1))`
`RandomApply([ColorJitter(0.4, 0.4, 0.4, 0.1)], p=0.1)`
`RandomGrayScale(p=0.2)`
`RandomHorizontalFlip()` |
| | Weak | `RandomResizedCrop(84, scale=(0.2, 1))`
`RandomHorizontalFlip()` |

**Training procedures.** To ensure the performance of PsCo and self-supervised learning models, we use three independently-trained models with random seeds and report the average performance of them.

## B  ANALYSIS ON HYPERPARAMETER SENSITIVITY

For the small-scale experiments, we use a momentum of $m = 0.99$ and a temperature of $\tau_{\text{PsCo}} = 1$. We here provide more ablation experiments with varying the hyperparameters $m$ and $\tau_{\text{PsCo}}$. Table 9 and 10 show the sensitivity of hyperparameters on the miniImageNet dataset. We observe that PsCo achieves good performance even for non-optimal hyperparameters.

Table 9: Sensitivity of momentum $m$ on mini-ImageNet (way, shot).

| $m$ | (5, 1) | (5, 5) | (5, 20) | (5, 50) |
|---|---|---|---|---|
| 0.9 | 46.49 | 62.18 | 70.21 | 72.77 |
| 0.99 | **46.70** | **63.26** | **72.22** | **73.50** |
| 0.999 | 45.96 | 61.53 | 69.66 | 72.04 |

Table 10: Sensitivity of temperature $\tau_{\text{PsCo}}$ on miniImageNet (way, shot).

| $\tau_{\text{PsCo}}$ | (5, 1) | (5, 5) | (5, 20) | (5, 50) |
|---|---|---|---|---|
| 0.2 | 46.43 | 62.29 | 70.04 | 72.22 |
| 0.5 | 46.32 | 62.63 | 70.50 | 73.15 |
| 1.0 | **46.70** | **63.26** | **72.22** | **73.50** |

## C    EFFECT OF ADAPTATION

We measure the performance with and without our adaptation scheme on various domains using miniImageNet-pretrained PsCo. Table 11 shows that our adaptation scheme enhances the way to adapt to each domain. In particular, the adaptation scheme is highly suggested for cross-domain few-shot classification scenarios.

Table 11: Before and after adaptation of PsCo in few-shot classification.

| Adaptation | miniImageNet | CUB | Cars | Places | Plantae | CropDiseases | EuroSAT | ISIC | ChestX |
|---|---|---|---|---|---|---|---|---|---|
| *5-way 5-shot* | | | | | | | | | |
| ✗ | 63.26 | 55.15 | 42.27 | 62.98 | 48.31 | 79.75 | 74.73 | 41.18 | 24.54 |
| ✓ | **63.30** | **57.38** | **44.01** | **63.60** | **52.72** | **88.24** | **81.08** | **44.00** | **24.78** |
| *5-way 20-shot* | | | | | | | | | |
| ✗ | 72.22 | 62.35 | 51.02 | 70.85 | 55.91 | 84.72 | 78.96 | 48.53 | 27.60 |
| ✓ | **73.00** | **68.58** | **57.50** | **73.95** | **64.53** | **94.95** | **87.65** | **54.59** | **27.69** |

## D    SETUP FOR STANDARD FEW-SHOT BENCHMARKS

We here describe details of benchmarks and baselines in Section D.1 and D.2, respectively, for the standard few-shot classification experiments (Section 4.1).

### D.1    DATASETS

**Omniglot** (Lake et al., 2011) is a $28 \times 28$ gray-scale dataset of 1623 characters with 20 samples each. We follow the setup of unsupervised meta-learning approaches (Hsu et al., 2019). We split the dataset into 120, 100, and 323 classes for meta-training, meta-validation, and meta-test respectively. In addition, the 0, 90, 180, and 270 degrees rotated views for each class become the different categories. Thus, we have a total of 6492, 400, and 1292 classes for meta-training, meta-validation, and meta-test respectively.

**MiniImageNet** (Ravi & Larochelle, 2017) is an $84 \times 84$ resized subset of ILSVRC-2012 (Deng et al., 2009) with 600 samples each. We split the dataset into 64, 16, and 20 classes for meta-training, meta-validation, and meta-test respectively as introduced in Ravi & Larochelle (2017).

### D.2    BASELINES

We compare our performance with unsupervised meta-learning, self-supervised learning, and supervised meta-learning methods. To be specific, (a) for the unsupervised meta-learning, we use CACTUs (Hsu et al., 2019) of the best options (ACAI clustering for Omniglot and DeepCluster for miniImageNet), UMTRA (Khodadadeh et al., 2019), LASIUM (Laskin et al., 2020) of the best options (LASIUM-RO-GAN for Omniglot and LASIUM-N-GAN for miniImageNet), Meta-GMVAE (Lee et al., 2021), Meta-SVEBM (Kong et al., 2021); (b) for the self-supervised learning methods, we use SimCLR (Chen et al., 2020a), MoCo v2 (Chen et al., 2020b), and SwAV (Caron et al., 2020); (c) for the supervised meta-learning, we use the results of MAML (Finn et al., 2017) and ProtoNets (Snell et al., 2017) reported in (Hsu et al., 2019).

For training self-supervised learning methods in our experimental setups, we use the same architecture and hyperparameters. For the hyperparameter of temperature scaling, we use the value provided in each paper: $\tau_{\texttt{SimCLR}} = 0.5$ for SimCLR, $\tau_{\texttt{MoCo}} = 0.2$ for MoCo v2, and $\tau_{\texttt{SwAV}} = 0.1$ for SwAV. For evaluation, we use K-Nearest Neightobrs (K-NN) for self-supervised learning methods since their classification rules are not specified.

# E    SETUP FOR CROSS-DOMAIN FEW-SHOT BENCHMARKS

We now describe the setup for cross-domain few-shot benchmarks, including detailed information on datasets, baseline experiments, implementational details, and the setup for large-scale experiments.

## E.1    DATASETS

For the cross-domain few-shot benchmarks, we use eight different datasets. We describe the dataset information in Table 12. We use the dataset split described in Tseng et al. (2020) for the benchmark of high-similarity and we use the dataset split described in Guo et al. (2020) for the benchmark of low-similarity. Because we do not perform the meta-training procedure using the datasets of cross-domain benchmarks, we only utilize the meta-test splits on these datasets. We use the $84 \times 84$ resized samples for evaluation on small-scale experiments.

Table 12: Information of datasets for cross-domain few-shot benchmarks.

| ImageNet similarity | Datset | # of classes | # of samples |
|---|---|---|---|
| High | CUB (Wah et al., 2011) | 200 | 11,788 |
| | Cars (Krause et al., 2013) | 196 | 16,185 |
| | Places (Zhou et al., 2018) | 365 | 1,800,000 |
| | Plantae (Horn et al., 2018) | 5089 | 675,170 |
| Low | CropDiseases (Mohanty et al., 2016) | 38 | 43,456 |
| | EuroSAT (Helber et al., 2019) | 10 | 27,000 |
| | ISIC (Codella et al., 2018) | 7 | 10,015 |
| | ChestX (Wang et al., 2017) | 7 | 25,848 |

## E.2    BASELINES

We compare our performance with (a) previous in-domain state-of-the-art methods of unsupervised meta-learning, self-supervised learning models, and supervised meta-learning models.

**Unsupervised meta-learning models.** We use previous in-domain state-of-the-art methods of unsupervised meta-learning models, Meta-GMVAE(Lee et al., 2021) and Meta-SVEBM (Kong et al., 2021). We use the miniImageNet pretrained parameters that the paper provided, and follow the meta-test procedure of each model to evaluate the performance.

**Self-supervised learning models.** We use SimCLR (Chen et al., 2020a), MoCo v2 (Chen et al., 2020b), and SwAV (Caron et al., 2020) of miniImageNet pretrained parameters as our baselines. Because self-supervised learning models are pretrained on miniImageNet, we additionally fine-tune the models with a linear classifier to let the models adapt to each domain. Following the setting provided in Guo et al. (2020); Oh et al. (2022), we detach the head of the models $g_\theta$ and attach the linear classifier $c_\psi$ to the model. We freeze the base network $f_\theta$ while fine-tuning and only $c_\psi$ is learned. We fine-tune the models via SGD with an initial learning rate of 0.01, a momentum of 0.9, weight decay of 0.001, and a batch size of $N = 4$ for 100 epochs.

**Supervised meta-learning models.** We use MAML (Finn et al., 2017) and ProtoNets (Snell et al., 2017) of Conv5 architectures of miniImageNet pretrained. Following the procedure of Snell et al. (2017), we train the models via Adam (Kingma & Ba, 2015) with a learning rate of 0.001 and cut the learning rate in half for every training of 2000 episodes. We train them for 60K episodes and use the model of the best validation accuracy. We train them through a 5-way 5-shot, and the rest of the hyperparameters are referenced in their respective papers. We observe that their performances are similar to the performance described in Table 1.

## E.3    EVALUATION DETAILS

To evaluate our method, we apply our adaptation scheme. Following Section 3.3, we freeze the base network $f_\theta$. We train only projection head $g_\theta$ and prediction head $h_\theta$ via SGD with an initial learning rate of 0.01, a momentum of 0.9, and weight decay of 0.001 as self-supervised learning models are fine-tuned. We only apply 50 iterations of our adaptation scheme when reporting performance.

### E.4 LARGE-SCALE SETUP

Here, we describe the setup for large-scale experiments. For evaluating, we use the same protocol with the small-scale experiments, except the scale of images is $224 \times 224$.

**Augmentations.** For large-scale experiments, we use $224 \times 224$-scaled data. Thus, we use similar yet slightly different augmentation schemes with small-scale experiments. Following the strong augmentation used in Chen et al. (2020b;a), we additionally apply `GaussianBlur` as a random augmentation. We use the same configuration for weak augmentation. For evaluation, we resize the images into $256 \times 256$ and then apply the `CenterCrop` to make $224 \times 224$ images by following Guo et al. (2020).

**ImageNet pretraining.** We pretrain MoCo v2 (Chen et al., 2020b), BYOL (Grill et al., 2020), and our PsCo of ResNet-18/50 (He et al., 2016) via SGD with a batch size of $N = 256$ for 200 epochs. Following (Chen et al., 2020b; Chen & He, 2021), we use an initial learning rate of 0.03 with the cosine learning schedule, $\tau_{\text{MoCo}} = 0.2$ and a weight decay of 0.0001. We use a queue size of $M = 65536$ and momentum of $m = 0.999$. For the parameters of PsCo, we use $\tau_{\text{PsCo}} = 0.2$ and $K = 16$ as the queue is 4 times bigger. For supervised pretraining, we use the the model checkpoint officially provided by torchvision (Paszke et al., 2019).

## F Experimental results with 95% confidence interval

We here provide the experimental results of Table 1, 2, and 3 with 95% confidence intervals in Table 13, 14, and 15, respectively.

Table 13: Few-shot classification accuracy (%) on Omniglot and miniImageNet with a 95% confidence interval over 2000 few-shot tasks.

| | Omniglot (way, shot) | | | | miniImageNet (way, shot) | | | |
|---|---|---|---|---|---|---|---|---|
| Method | (5, 1) | (5, 5) | (20, 1) | (20, 5) | (5, 1) | (5, 5) | (5, 20) | (5, 50) |
| SimCLR | 92.13±0.30 | 97.06±0.13 | 80.95±0.21 | 91.60±0.12 | 43.35±0.42 | 52.50±0.39 | 61.83±0.35 | 64.85±0.32 |
| MoCo v2 | 92.66±0.28 | 97.38±0.12 | 82.13±0.21 | 92.34±0.11 | 41.92±0.41 | 50.94±0.38 | 60.23±0.35 | 63.45±0.33 |
| SwAV | 93.13±0.27 | 97.32±0.13 | 82.63±0.21 | 92.12±0.12 | 43.24±0.42 | 52.41±0.39 | 61.36±0.35 | 64.52±0.33 |
| PsCo (ours) | **96.37**±0.20 | **99.13**±0.07 | **89.60**±0.17 | **97.07**±0.07 | **46.70**±0.42 | **63.26**±0.37 | **72.22**±0.32 | **73.50**±0.29 |

Table 14: Few-shot classification accuracy (%) on cross-domain few-shot classification benchmarks of Conv5 pretrained on miniImageNet with a 95% confidence interval over 2000 few-shot tasks.

(a) Cross-domain few-shot benchmarks similar to miniImageNet.

| | CUB | | Cars | | Places | | Plantae | |
|---|---|---|---|---|---|---|---|---|
| Method | (5, 5) | (5, 20) | (5, 5) | (5, 20) | (5, 5) | (5, 20) | (5, 5) | (5, 20) |
| Meta-GMVAE | 47.48±0.47 | 54.08±0.45 | 31.39±0.34 | 38.36±0.35 | 57.70±0.47 | 65.08±0.38 | 38.27±0.40 | 45.02±0.37 |
| Meta-SVEBM | 45.50±0.83 | 54.61±0.91 | 34.27±0.79 | 46.23±0.87 | 51.27±0.82 | 61.09±0.85 | 38.12±0.86 | 46.22±0.85 |
| SimCLR | 52.11±0.45 | 61.89±0.45 | 37.40±0.35 | 50.05±0.39 | 60.10±0.40 | 69.93±0.35 | 43.42±0.37 | 54.92±0.36 |
| MoCo v2 | 53.23±0.45 | 62.81±0.45 | 38.65±0.35 | 51.77±0.39 | 59.09±0.40 | 69.08±0.36 | 43.97±0.37 | 55.45±0.36 |
| SwAV | 51.58±0.45 | 61.38±0.46 | 36.85±0.33 | 50.03±0.38 | 59.57±0.40 | 69.70±0.36 | 42.68±0.37 | 54.03±0.36 |
| PsCo (ours) | **57.38**±0.44 | **68.58**±0.41 | **44.01**±0.39 | **57.50**±0.40 | **63.60**±0.41 | **73.95**±0.36 | **52.72**±0.39 | **64.53**±0.36 |
| *MAML* | 56.57±0.43 | 64.17±0.40 | 41.17±0.40 | 48.82±0.40 | 60.05±0.42 | 67.54±0.37 | 47.33±0.41 | 54.86±0.38 |
| *ProtoNets* | 56.74±0.43 | 65.03±0.41 | 38.98±0.37 | 47.98±0.38 | 59.39±0.40 | 67.77±0.36 | 45.89±0.40 | 54.29±0.38 |

(b) Cross-domain few-shot benchmarks dissimilar to miniImageNet.

| | CropDiseases | | EuroSAT | | ISIC | | ChestX | |
|---|---|---|---|---|---|---|---|---|
| Method | (5, 5) | (5, 20) | (5, 5) | (5, 20) | (5, 5) | (5, 20) | (5, 5) | (5, 20) |
| Meta-GMVAE | 73.56±0.53 | 81.22±0.39 | 73.83±0.42 | 80.11±0.35 | 33.48±0.30 | 39.48±0.28 | 23.23±0.23 | 26.26±0.24 |
| Meta-SVEBM | 71.82±1.03 | 83.13±0.78 | 70.83±0.83 | 80.21±0.73 | 38.85±0.76 | 48.43±0.81 | **26.26**±0.65 | 28.91±0.69 |
| SimCLR | 79.90±0.39 | 88.73±0.28 | 79.14±0.38 | 85.05±0.32 | 42.83±0.29 | 51.35±0.27 | 25.14±0.23 | **29.21**±0.24 |
| MoCo v2 | 80.96±0.37 | 89.85±0.27 | 79.94±0.37 | 86.16±0.31 | 43.43±0.30 | 52.14±0.27 | 25.24±0.23 | 29.19±0.24 |
| SwAV | 80.15±0.39 | 89.24±0.28 | 79.31±0.39 | 85.62±0.31 | 43.21±0.30 | 51.99±0.27 | 24.99±0.23 | 28.57±0.24 |
| PsCo (ours) | **88.24**±0.31 | **94.95**±0.18 | **81.08**±0.35 | **87.65**±0.28 | **44.00**±0.30 | **54.59**±0.29 | 24.78±0.23 | 27.69±0.23 |
| *MAML* | 77.76±0.39 | 83.24±0.34 | 71.48±0.38 | 76.70±0.33 | 47.34±0.37 | 55.09±0.34 | 22.61±0.22 | 24.25±0.22 |
| *ProtoNets* | 76.01±0.40 | 83.64±0.33 | 64.91±0.38 | 70.88±0.33 | 40.62±0.31 | 48.38±0.29 | 23.15±0.22 | 25.72±0.23 |

Table 15: Few-shot classification accuracy (%) on cross-domain few-shot classification benchmarks of pretrained ResNet-18/50 on ImageNet with a 95% confidence interval (5-way 5-shot).

| Methods | CUB | Cars | Places | Plantae | CropDiseases | EuroSAT | ISIC | ChestX |
|---|---|---|---|---|---|---|---|---|
| *ResNet-18 pretrained* | | | | | | | | |
| MoCo v2 | 61.88±0.96 | 46.42±0.73 | 79.11±0.68 | 56.24±0.72 | 81.48±0.74 | 75.98±0.73 | 38.21±0.53 | 24.34±0.36 |
| +PsCo (Ours) | **70.08**±0.87 | **50.73**±0.76 | **79.74**±0.64 | **61.55**±0.76 | **87.91**±0.57 | **79.92**±0.64 | **40.61**±0.52 | **25.03**±0.42 |
| *ResNet-50 pretrained* | | | | | | | | |
| MoCo v2 | 64.16±0.91 | 47.67±0.75 | 81.39±0.64 | 61.36±0.79 | 82.89±0.77 | 76.96±0.68 | 38.26±0.56 | **24.28**±0.39 |
| +PsCo (Ours) | **76.63**±0.84 | **53.45**±0.76 | **83.87**±0.58 | **69.17**±0.70 | **89.85**±0.78 | **83.99**±0.52 | **41.64**±0.55 | 23.60±0.36 |
| BYOL | 67.45±0.88 | 45.74±0.76 | 75.43±0.79 | 56.86±0.84 | 80.82±0.86 | 77.70±0.71 | 37.27±0.56 | 24.15±0.36 |
| +PsCo (Ours) | **82.13**±0.70 | **56.19**±0.76 | **83.80**±0.62 | **71.14**±0.71 | **92.92**±0.44 | **85.33**±0.54 | **42.90**±0.55 | **26.05**±0.46 |
| Supervised | 89.13±0.55 | 75.15±0.75 | 84.41±0.61 | 72.91±0.73 | 90.96±0.48 | 85.64±0.52 | 43.34±0.57 | 25.35±0.41 |

