# OpenReview forum: "Unsupervised Meta-learning via Few-shot Pseudo-supervised Contrastive Learning"
_ICLR.cc/2023/Conference — ICLR 2023 notable top 25%_

### Official Review · Reviewer_MCTB · 2022-10-23

**Confidence:** 3
**Correctness:** 3
**Technical Novelty And Significance:** 3
**Empirical Novelty And Significance:** Not applicable
**Recommendation:** 8

**Clarity, Quality, Novelty And Reproducibility:**

The paper is of high quality but some aspects need to be improved in the response (see weaknesses).  The idea is original and novel and the extensive experiments demonstrate its effectiveness. The authors need to release the code to make their work reproducible.

**Strength And Weaknesses:**

Strengths:
- The paper is well organized and well written.
- The idea of progressively improving pseudo-labelling strategy  during meta-learning is novel. Inspired by self-supervised learning, authors achieve this using momentum queue and momentum network. Since the shots are selected from the entire dataset, the tasks constructed in the proposed approach are diverse.
- The experiments are comprehensive and the method outperforms other unsupervised meta-learning methods.
- The method scales to large datasets better than existing baselines.

Weaknesses:
- When constructing the pseudo-tasks, the method first randomly selects N samples and treats them as N queries of different N labels. What if selected samples share the same label, but we assume that they have different labels and try to separate them in the feature space during training? This needs to be clarified.
- The method constructs the pseudo-tasks using the queue Q of previous mini-batch samples. Where does the Q come from for the first few batches? It would be more clear if there is more explanation about pseudo-tasks construction.
- The authors compare only to MAML and ProtoNet as supervised meta-learning benchmarks which are not SOA methods so the statements about outperforming supervised meta-learning benchmarks need to be toned-down. Furthermore, for cross-domain learning how would MAML and ProtoNet perform if the representations were pretrained using self-supervised learning before? It is contra-intuitive that unsupervised meta-learning outperform supervised meta-learning and the authors need to be careful about such statements.
- Why do authors compare only to MoCO on Image-Net and not to other self-supervised learning baselines such as SimCLR and RotationNet?
- The authors claim that their method is more efficient than other unsupervised meta-learning baselines but they do not compare running time of their approach and the baselines. Additional information that would be needed is the comparison of running time of their approach and self-supervised learning baselines on the ImageNet.



**Summary Of The Paper:**

The authors propose Pseudo-supervised Contrast (PsCo), an unsupervised meta-learning framework that solves a few-shot learning problem from the constructed few-shot pseudo-tasks. The key idea of the approach is to improve pseudo-labelling progressively during meta-learning by using current and previous mini-batches which is achieved using momentum network and momentum queue. In particular, few-shot tasks are constructed using the current mini-batch to select queries and the queue of previous mini-batches to select K shots. The label assignment matrix is found using Sinkhorn-Knopp algorithm. The proposed approach is evaluated on in-domain and cross-domain few-shot classification benchmarks and the method is shown to outperform baselines.


**Summary Of The Review:**

Overall, this is a good paper with an interesting approach and well designed experiments. There are some things that should be improved and some details remain unclear or are not convincing. The authors should clarify these points in their response.

---

> ### Author Response · Authors · 2022-11-18
> **Response to Reviewer MCTB (2/2)**
>
> **[Q4]** Why do authors compare only to MoCo on ImageNet and not to others e.g., SimCLR, RotationNet? \
> **[A4]** This is because our PsCo framework is built on top of MoCo. In Table 3, we aim to verify that PsCo can improve the self-supervised learning baseline (i.e., MoCo) under the large-scale setting. At the time of submission, we excluded other self-supervised learning methods due to our limited computational resources; in principle, PsCO can be incorporated with any self-supervised learning scheme to improve its performance.
>
> To address your concern, we additionally conduct ImageNet experiments during this rebuttal period with two transfer learning baselines (see the below table or Table 3 in the revision): (a) a self-supervised learning method, BYOL [1], and (b) the supervised off-the-shelf ImageNet-pretrained model (officially provided by torchvision). Note that BYOL [1] is non-contrastive unlike MoCo and it has been widely utilized across various domains (e.g., graph [2]) with strong performance [3]. We also verify that PsCo can be incorporated with BYOL instead of MoCo. As shown in the below table, our PsCo objective significantly improves the non-contrastive approach, BYOL. Furthermore, PsCo + BYOL achieves comparable performance (except Cars) with the supervised off-the-shelf ImageNet-pretrained model. These results further highlight our strength: high compatibility with various self-supervised learning methods.
>
> \begin{array}{l|ccccccccc}
> \hline
>        & \text{CUB} & \text{Cars} & \text{Places} & \text{Plantae} & \text{CropDisases} & \text{EuroSAT} & \text{ISIC} & \text{ChestX} \newline
> \hline
> \text{MoCo}        & 64.16 & 47.67 & 81.39 & 61.36 & 82.89 & 76.96 & 38.26 & \bf{24.28}  \newline
> \text{MoCo + PsCo (Ours)} & \bf{76.63} & \bf{53.45} & \bf{83.87} & \bf{69.17} & \bf{89.85} & \bf{83.99} & \bf{41.64} & 23.60 \newline
> \hline
> \text{BYOL}        & 67.45 & 45.74 & 75.43 & 56.86 & 80.82 & 77.70 & 37.27 & 24.15 \newline
> \text{BYOL + PsCo (Ours)} & \bf{82.13} & \bf{56.19} & \bf{83.80} & \bf{71.14} & \bf{92.92} & \bf{85.33} & \bf{42.90} & \bf{26.05} \newline
> \hline
> \text{Supervised}  & 89.13 & 75.15 & 84.41 & 72.91 & 90.96 & 85.64 & 43.34 & 25.35
> \newline
> \hline
> \end{array}
>
> [1] Grill et al., Bootstrap Your Own Latent: A New Approach to Self-supervised Learning, NeurIPS 2020 \
> [2] Thakoor et al., Large-Scale Representation Learning on Graphs via Bootstrapping, ICLR 2022 \
> [3] Ericsson et al., How Well Do Self-Supervised Models Transfer?, CVPR 2021
>
> ---
>
> **[Q5]** Comparison of running time \
> **[A5]** Thank you for the constructive feedback. As you suggested, we compare the running times of our method, PsCo, and the best baseline, Meta-SVEBM, on mini-ImageNet. As a result, we observe that PsCo is 2.7x faster than Meta-SVEBM. This could be critical in a large-scale benchmark: e.g., PsCo takes 4 days for ImageNet training, which means Meta-SVEBM will take 11 days (in expectation). We included this discussion in the revision (Section 4.2).

---

> ### Author Response · Authors · 2022-11-18
> **Response to Reviewer MCTB (1/2)**
>
> Dear Reviewer MCTB,
>
> We sincerely thank you for your helpful feedback and insightful comments. We address your comments and questions below.
>
> ---
>
> **[Q1]** What if randomly selected samples share the same label but assume they have different labels and separate them in the feature space? \
> **[A1]** Thank you for your constructive question. The case you mentioned is not critical in contrastive learning in general, as well as our PsCo framework. This is because even if randomly selected samples could share the same label, the samples often contain different features (e.g., background, color, shape), hence, separating them could be beneficial. This has been well verified in the self-supervised learning literature [1].
>
> [1] Ericsson et al., How Well Do Self-supervised Models Transfer?, CVPR 2021
>
> ---
>
> **[Q2]** Where does the Q come from for the first few batches? \
> **[A2]** Thank you very much for the question. We randomly initialize the queue Q at the beginning of meta-training as in [1]. Hence, we use random representations for the first few batches. We clarified this part in the final draft (Section 3.1).
>
> [1] He et al., Momentum Contrast for Unsupervised Visual Representation Learning, CVPR 2020
>
> ---
>
> **[Q3]** “Outperforming supervised meta-learning” needs to be toned-down. It is contra-intuitive that unsupervised outperform supervised setting. \
> **[A3]** Thank you for the suggestion to improve our manuscript. We tone-downed the statement in the revision (Section 1 and 4.2).

---

### Official Review · Reviewer_3zZw · 2022-10-25

**Confidence:** 4
**Correctness:** 4
**Technical Novelty And Significance:** 2
**Empirical Novelty And Significance:** 3
**Recommendation:** 8

**Clarity, Quality, Novelty And Reproducibility:**

The authors mention that they will release their code upon acceptance. The pseudo code of algorithm is provided and the details are explained to a good degree.

**Strength And Weaknesses:**

Strength:
Interesting and novel algorithm.
Good and structured writing and presentation.
Comparing with cross-domain meta-learning.


**Summary Of The Paper:**

The authors propose a novel algorithm for unsupervised few-shot meta-learning that combines contrastive learning with unsupervised task generation to train a few-shot model. They achieve good improvements on state-of-the-art benchmarks as well as cross-domain meta-learning benchmarks.

**Summary Of The Review:**

Interesting and novel paper. Given the strength and reproducibility comments I made, I tend to vote for acceptance of the paper.

---

> ### Author Response · Authors · 2022-11-18
> **Response to Reviewer 3zZw**
>
> Dear Reviewer 3zZw,
>
> We sincerely thank you for your positive feedback on our paper! If you have any questions or suggestions, please let us know!
>
> Best, \
> Authors

---

### Official Review · Reviewer_yfDU · 2022-10-25

**Confidence:** 4
**Correctness:** 3
**Technical Novelty And Significance:** 3
**Empirical Novelty And Significance:** 3
**Recommendation:** 6

**Clarity, Quality, Novelty And Reproducibility:**

Clarity
=====
The paper is clear for the most part, with a few exceptions below.

In Equation 1, state what M represents exactly.

In Equation 1, for negative pairs where A_{i,j} is 0, the term inside the log will be 0, so the contrastive loss in this case will not be a function of the similarity between q_i and k_j. Is this an error? E.g. should A_{i,j} be -1 instead of 0 for negative pairs?

Notation: k_i (e.g. in Equation 2) and z_i (in Section 3.1) seem to refer to the same quantity. Am I missing something? If not, change the notation to be consistent.

In Table 4, it’s not clear what removing Sinkhorn entails exactly. What is done instead? I’m assuming some different matching algorithm is used?

Novelty
======
While the proposed method heavily relies on MoCo, the particular adaptation to the problem of unsupervised meta-learning is novel to the best of my knowledge and interesting.

Reproducibility
============
The authors discuss implementation details in the Appendix. It’s hard to say how easy it would be to reproduce the presented results. I recommend the authors to release code upon publication.

Quality
======
The technical quality of the paper is high for the most part. However, I have some concerns about experimental comparisons, ablations.

Were SimCLR / MoCo trained on the meta-training set of Omniglot / mini-ImageNet for the experiments? This is kind of strange as the strength of these approaches exhibits itself when training on large datasets, with architectures much larger than 4 or 5 convolutional layers. So I think to truly compare against those approaches, it would be more interesting to use larger-scale pre-training.

The supervised meta-learning baselines used here are old and really far from the state-of-the-art, especially for cross-domain few-shot learning.

In Table 3, why are comparisons made only against MoCo v2? For instance, in addition to the self-supervised and supervised meta-learning baselines, it would be interesting to compare to a supervised off-the-shelf ImageNet trained model (that was not meta-learned, was simply trained for supervised classification), by finetuning it on each transfer task.

More generally, add transfer learning baselines to the paper. In the scenario of small architectures (4 or 5 conv layers), transfer learning is likely to not work very well, but for larger architectures and more training data it seems to dominate.

Missing ablation: train with only L_{PsCo} instead of the sum of that loss with MoCo’s loss; that is, train only for the contrastive loss that matches each query (example in the current batch) with the corresponding selected ‘support examples’ from the queue (and pushes it away from other support examples that were assigned different pseudolabels).


**Strength And Weaknesses:**

Strengths
========
[+] This is a really neat idea for leveraging progress in self-supervised learning for the problem of unsupervised meta-learning

[+] The paper is for the most part clearly written and well-organized

[+] Empirically, the proposed method seems to outperform some previous baselines

[+] A thorough ablation analysis is presented along with a set of additional experiments to further understand the performance of the proposed method

Weaknesses
===========
[-] The problem of unsupervised meta-learning is poorly motivated. More generally, whether meta-learning is the most fruitful approach for few-shot classification is questionable. Recent works [1-3] (see references below) show that simpler methods often work better. So it’s not clear why this framework is useful.

[-] The empirical investigation is limited to simple benchmarks for the most part (though I appreciated the cross-domain experiments), limited to very small convnet architectures (the few-shot learning community has long ago started using resnet-12 [1], resnet-18 [4], and larger models [3]), and the baselines compared against for supervised meta-learning are very basic and very far from the state-of-the-art.

References
==========
[1] A closer look at few-shot classification. Chen et al. ICLR 2019.

[2] Revisiting few-shot image classification: A good embedding is all you need? Tian et al. 2020.

[3] Comparing Transfer and Meta Learning Approaches on a Unified Few-Shot Classification Benchmark. Dumoulin et al. NeurIPS 2021.

[4] Meta-Dataset: A Dataset of Datasets for Learning to Learn from Few Examples. Triantafillou et al. ICLR 2020.


**Summary Of The Paper:**

This paper proposes a new approach to unsupervised meta-learning by drawing inspiration from recent self-supervised learning methods. Specifically, they employ an architecture similar to MoCo, where one branch is updated with momentum updates of the other branch, which is updated ‘online’. To form each task for meta-learning, they treat each example in the current batch as a ‘query’ and then propose a way of selecting relevant ‘shots’ for each query from a queue, where (the representation of) examples seen in previous batches are added, to serve as the support set for the episode. They utilize the Sinkhorn-Knopp algorithm to do this matching and pick the top K examples from the queue, for each query (where K is the shot of each class). Constructing tasks dynamically in this way can be advantageous compared to the approach followed by previous works where the set of tasks for meta-learning is created at the start of the meta-training phase and fixed, as this can lead to more variety as well as ‘better’ tasks (e.g. the support set of a class is a good match for its query set), since the representations become more informative throughout training. The use of a queue not only is useful for training with the MoCo objective in a self-supervised manner, but also can increase diversity within the episode, since the support examples can be selected from previous mini-batches too, leading to a larger pool of candidates. They use a supervised contrastive algorithm to solve each meta-training task. Their overall loss for training is this per-task contrastive loss, as well as the usual MoCo loss. At meta-test time, they use a nearest-centroid algorithm for single-source few-shot learning, while for cross-domain scenarios they further adapt the projection layers (small number of parameters) to handle the distribution shift. Specifically, they adapt on the support set via supervised contrastive learning where each support example plays the role of a ‘query’ and the pool for ‘searching’ is the rest of the support set; positive examples are those with the same label, since labels are known for the support examples. Empirically, they show improved results on some benchmarks over some previous methods.

**Summary Of The Review:**

Overall, I found the idea presented in this paper interesting, the paper well-written and well-organized. I’m generally unconvinced of the value of unsupervised meta-learning, or meta-learning in general as a means for representation learning for the problem of few-shot classification, especially lacking transfer learning baselines. I also found the experimental setup to be not very convincing, as it focuses on very small architectures and the comparisons are against models that are far from state-of-the-art (for the supervised meta-learning baselines). These are the reasons that I recommend rejection.

---

> ### Author Response · Authors · 2022-11-18
> **Response to Reviewer yfDU (3/3)**
>
> **[Q9]** Missing ablation: train with only $\mathcal{L}_\text{PsCo}$ \
> **[A9]** Thank you for the constructive suggestion. The below table shows the suggested ablation experiment. We found that training with only $\mathcal{L}_\text{PsCo}$ achieves moderate performance. We think that $\mathcal{L}_\text{MoCo}$ helps to construct semantically meaningful few-shot tasks via better representations. We included the result into the revision (Table 4).
> \begin{array}{l|cccc}
> \hline
> & \text{5-way 1-shot} & \text{5-way 5-shot} & \text{20-way 1-shot} & \text{20-way 5-shot} \newline
> \hline
> \mathcal{L}_\mathtt{PsCo}+\mathcal{L}_\mathtt{MoCo} & 96.37 & 99.13 & 89.64 & 97.07 \newline
> \mathcal{L}_\mathtt{PsCo} & 93.16 & 97.40 & 81.03 & 91.33 \newline
> \hline
> \end{array}

---

> > ### Comment · Reviewer_yfDU · 2022-12-06
> > **thank you for the thorough rebuttal**
> >
> > Hi authors,
> >
> > Thank you for the thorough responses and additional experiments. I think the additional ablation and comparisons against stronger transfer learning baselines, as well as the clarifications to the paper strengthen the submission significantly. I also like the discussion about these results providing a data point about the role that meta-learning may play for few-shot learning and how it may complement transfer learning solutions. In light of these modifications, I raise my score to an accept.

---

> ### Author Response · Authors · 2022-11-18
> **Response to Reviewer yfDU (2/3)**
>
> **[Q3]** Clarification about Equation 1: (1) meaning of M, and (2) the term inside the log can be 0 when $A_{i,j}$ is 0. \
> **[A3-1]** “M” in Equation 1 refers to the number of key samples for contrastive learning. For example, M can be the queue size of the momentum queue for MoCo. \
> **[A3-2]** Thank you very much for pointing this out! We fixed the typo in Equation 1.
>
> ---
> **[Q4]** Clarification about $k_i$ of Equation 2 and $z_i$ of Section 3.1. \
> **[A4]** We would like to clarify that these two values do not indicate the same quantity. We write $k_i$ for the general use of keys for contrastive learning, while $z_i$ indicates the representations of the current mini-batch. Similarly, $\tilde{z}_i$ means the momentum representations of previous mini-batches in the queue. In this sense, $k_i$ can be used as {$\tilde{z}_i$}, or {$z_i, \tilde{z}_i$} as described in Section 3.1.
>
> ---
>
> **[Q5]** Clarification about Table 4: meaning of removing Sinkhorn. \
> **[A5]** “Removing Sinkhorn” in Table 4 refers to using the similarity matrix $(z_i^\top \tilde{z}_j)$ as an assignment matrix $\tilde{A}$ directly without solving the optimization problem (4). Without solving the problem via the Sinkhorn algorithm, the support samples (i.e., shots) obtained by the top-K selection could be overlapped as shown in Figure 2(b).
>
> ---
>
> **[Q6]** Reproducibility of code \
> **[A6]** We remark that our code was already attached in our supplementary material. We also plan to publicly release our code once the decision is made.
>
> ---
>
> **[Q7]** Were SimCLR / MoCo trained on the meta-training set of Omniglot / mini-ImageNet for the experiments? It would be more interesting to use larger-scale pre-training. \
> **[A7]** Yes for only Table 1 and 2: all the methods in the tables are trained on either Omniglot or mini-ImageNet dataset for the fair comparison. As you suggested, we already provided large-scale pre-training experiments in Table 3. Specifically, we performed training ResNet-50 on ImageNet, and then evaluated the models under the cross-domain benchmarks.
>
> ---
>
> **[Q8]** Why are comparisons made only against MoCo v2 in Table 3? Add transfer learning baselines. \
> **[A8]** This is because our PsCo framework is built on top of MoCo. In Table 3, we aim to verify that PsCo can improve the self-supervised learning baseline (i.e., MoCo) under the large-scale setting. At the time of submission, we excluded other self-supervised learning methods due to our limited computational resources; in principle, PsCo can be incorporated with any self-supervised learning scheme to improve its performance.
>
> To address your concern, we additionally conduct ImageNet experiments during this rebuttal period with two transfer learning baselines (see the below table or Table 3 in the revision): (a) a self-supervised learning method, BYOL [1], and (b) the supervised off-the-shelf ImageNet-pretrained model (officially provided by torchvision). Note that BYOL [1] is non-contrastive unlike MoCo and it has been widely utilized across various domains (e.g., graph [2]) with strong performance [3]. We also verify that PsCo can be incorporated with BYOL instead of MoCo. As shown in the below table, our PsCo objective significantly improves the non-contrastive approach, BYOL. Furthermore, PsCo + BYOL achieves comparable performance (except Cars) with the supervised off-the-shelf ImageNet-pretrained model. These results further highlight our strength: high compatibility with various self-supervised learning methods.
> \begin{array}{l|ccccccccc}
> \hline
>        & \text{CUB} & \text{Cars} & \text{Places} & \text{Plantae} & \text{CropDisases} & \text{EuroSAT} & \text{ISIC} & \text{ChestX} \newline
> \hline
> \text{MoCo}        & 64.16 & 47.67 & 81.39 & 61.36 & 82.89 & 76.96 & 38.26 & \bf{24.28}  \newline
> \text{MoCo + PsCo (Ours)} & \bf{76.63} & \bf{53.45} & \bf{83.87} & \bf{69.17} & \bf{89.85} & \bf{83.99} & \bf{41.64} & 23.60 \newline
> \hline
> \text{BYOL}        & 67.45 & 45.74 & 75.43 & 56.86 & 80.82 & 77.70 & 37.27 & 24.15 \newline
> \text{BYOL + PsCo (Ours)} & \bf{82.13} & \bf{56.19} & \bf{83.80} & \bf{71.14} & \bf{92.92} & \bf{85.33} & \bf{42.90} & \bf{26.05} \newline
> \hline
> \text{Supervised}  & 89.13 & 75.15 & 84.41 & 72.91 & 90.96 & 85.64 & 43.34 & 25.35
> \newline
> \hline
> \end{array}
>
> [1] Grill et al., Bootstrap Your Own Latent: A New Approach to Self-supervised Learning, NeurIPS 2020 \
> [2] Thakoor et al., Large-Scale Representation Learning on Graphs via Bootstrapping, ICLR 2022 \
> [3] Ericsson et al., How Well Do Self-Supervised Models Transfer?, CVPR 2021

---

> ### Author Response · Authors · 2022-11-18
> **Response to Reviewer yfDU (1/3)**
>
> Dear Reviewer yfDU,
>
> We sincerely thank you for your helpful feedback and insightful comments. We address your comments and questions below.
>
> ---
>
> **[Q1]** Whether meta-learning is the most fruitful approach for few-shot classification is questionable. Recent works [1-3] show that simpler methods often work better. \
> **[A1]** We politely disagree with the reviewer’s opinion. A recent work [4], published concurrently with [3], shows that meta-learning is still effective over simple transfer methods [1-2] for few-shot classification, and the recent meta-learning literature has shown state-of-the-art performance in the few-shot problem [5-7]. We would like to emphasize that meta-learning for the few-shot classification problem is still an ongoing research topic. Furthermore, [8] shows that the meta-learning framework can improve self-supervised learning under various transfer learning scenarios. Overall, we strongly believe that the value of meta-learning should not be overlooked and our work could be a meaningful step toward bridging the gap between meta-learning and self-supervised learning.
>
> [1] Chen et al., A closer look at few-shot classification, ICLR 2019 \
> [2] Tian et al., Revisiting few-shot image classification: A good embedding is all you need?, 2020 \
> [3] Dumoulin et al., Comparing Transfer and Meta Learning Approaches on a Unified Few-Shot Classification Benchmark, NeurIPS 2021 \
> [4] Chen et al., Meta-Baseline: Exploring Simple Meta-Learning for Few-Shot Learning, ICCV 2021 \
> [5] Kang et al., Relational Embedding for Few-Shot Classification, ICCV 2021 \
> [6] Wu et al., Task-aware Part Mining Network for Few-Shot Learning, ICCV 2021 \
> [7] Chikontwe et al., CAD: Co-Adapting Discriminative Features for Improved Few-Shot Classification, CVPR 2022 \
> [8] Ni et al., The Close Relationship Between Contrastive Learning and Meta-Learning, ICLR 2022
>
> ---
>
> **[Q2]** The empirical investigation is limited to simple benchmarks for the most part \
> **[Q2-1]** Limited to very small ConvNet architectures \
> **[A2-1]** As you mentioned, our scheme is more favorable to work under larger architectures as it utilizes self-supervised learning techniques (which are known to work better under larger architectures). Namely, as reported in Table 3, we already verified the effectiveness of our scheme under large-scale cross-domain benchmarks with **a large architecture (e.g., ResNet50) and a large dataset (e.g., ImageNet)**. However, in Table 1-2, we follow **the standard benchmark setup commonly used in the unsupervised meta-learning literature** [1-5] for fair and reproducible comparison. We strongly believe that the scalability to larger architectures and datasets is rather the strength of our method, compared to recent unsupervised meta-learning schemes relying on generative models.
>
> **[Q2-2]** Supervised meta-learning methods are very basic and very far from the state-of-the-art \
> **[A2-2]** In this work, **we aim at unsupervised meta-learning, not supervised meta-learning**. Namely, supervised meta-learning methods are not important baselines for us. Therefore, we think the comparison with simple supervised baselines is enough just for the readers’ information, and also the standard choice in both unsupervised meta-learning [1-5] and self-supervised learning literature [6-8].
>
> Nevertheless, to alleviate your concern, we additionally conduct ResNet-12 experiments: we train our PsCo on mini-ImageNet (without labels) using the ResNet-12 architecture, as you suggested. In the setup, our PsCo model achieves 79.71% accuracy under 5-way 5-shot classification. This result is still comparable with the recent supervised meta-learning methods, e.g., Meta-Baseline [9] (79.26%) and ReNet [10] (82.58%). We believe this result further highlights the strength of our method.
>
> [1] Hsu et al., Unsupervised learning via Meta-learning, ICLR 2019 \
> [2] Khodadadeh et al., Unsupervised Meta-learning for Few-shot Image Classification, NeurIPS 2019 \
> [3] Khodadadeh et al., Unsupervised Meta-learning through Latent-space Interpolation in Generative Models, ICLR 2021 \
> [4] Lee et al., Meta-GMVAE: Mixture of Gaussian VAE for Unsupervised Meta-learning, ICLR 2021 \
> [5] Kong et al., Unsupervised Meta-learning via Latent Space Energy-based Model of Symbol Vector Coupling, NeurIPSW-MetaLearn 2021 \
> [6] Chen et al., A Simple Framework for Contrastive Learning of Visual Representations, ICML 2020 \
> [7] Grill et al., Bootstrap Your Own Latent: A New Approach to Self-supervised Learning, NeurIPS 2020 \
> [8] Caron et al., Emerging Properties in Self-Supervised Vision Transformers, ICCV 2021 \
> [9] Chen et al., Meta-Baseline: Exploring Simple Meta-learning for Few-shot Learning, ICCV 2021\
> [10] Kang et al, Relational Embedding for Few-shot Classification, ICCV 2021

---

### Official Review · Reviewer_eLS6 · 2022-10-27

**Confidence:** 5
**Correctness:** 4
**Technical Novelty And Significance:** 4
**Empirical Novelty And Significance:** 4
**Recommendation:** 8

**Clarity, Quality, Novelty And Reproducibility:**

Clarity

The paper is well written and easy to understand.

Reproducibility
The technique is certainly described in enough detail to be reproducible.

Quality and Novelty

The paper advances significantly over the state of the art in both the algorithm and the experimental results. The proposed technique is novel.

**Strength And Weaknesses:**

Strengths
1. Sound motivation of the work through thorough literature survey.
2. Innovative progressive refinement of pseudo-labels through the combination of the momentum network and a batch queue that ensures that information from past batches is used throughout.
3. Sound experimental results that show a clear advance over the state of the art.

Weaknesses

1. No significant weaknesses.

**Summary Of The Paper:**

This paper proposes a new approach to meta-learning that addresses vulnerability to pseudo-labeling errors in state of the art meta-learning methods by improving pseudo-labeling using a momentum network and a queue of previous batches. The main insight in this work comes from the recognition that pseudo-labels are usually fixed in state of the art methods and not allowed to change as new information comes in. The proposed technique enables changing of the pseudo-labels using cross batch training. The proposed technique advances over the state the state of the art significantly when applied to a variety of few-shot classification benchmark tests. The proposed technique also advances over the state of the art in its scalability.



**Summary Of The Review:**

The paper motivates the proposed technique well. The proposed technique is sound and innovative, with a strong main insight of adapting pseudo-labels progressively. That represents a clear advance over the state of the art. The experimental results also show advancement over the state of the art. Good paper.

---

> ### Author Response · Authors · 2022-11-18
> **Response to Reviewer eLS6**
>
> Dear eLS6,
>
> We sincerely thank you for your positive feedback on our paper! If you have any questions or suggestions, please let us know!
>
> Best, \
> Authors

---

### Author Response · Authors · 2022-11-18
**General Response**

Dear reviewers and AC,

We sincerely appreciate your valuable time and effort spent reviewing our manuscript.

As reviewers highlighted, we propose an interesting and novel method (ALL Reviewers) with strong empirical results (Reviewer eLS6, yfDU, MCTB) and extensive experiments (Reviewer 3zZw, MCTB). Our paper is also well-motivated (Reviewer eLS6) and well-written (Reviewer yfDU, 3zZw, MCTB).

We appreciate your constructive comments on our manuscript. In response to the comments, we have carefully revised and enhanced the manuscript with the following additional discussions and experiments:

- Add more ImageNet experiments with self-supervised and supervised baselines (Table 3)
- Add an ablation experiment (Table 4)
- Add discussion about efficiency (Section 4.2)
- Clarify descriptions and statements throughout our manuscript

These updates are temporarily highlighted in “$\color{blue}\text{blue}$” for your convenience to check.

We hope our response and revision sincerely address all the reviewers’ concerns.

Thank you very much.

Best regards, \
Authors.

---

### Decision · Program_Chairs · 2023-01-20

**Decision:**

Accept: notable-top-25%

**Justification For Why Not Higher Score:**

The presented method's scope is still limited, with a limited and fairly "artificial kinda" of setting. I do not think it is at an oral level, which is supposed to interest a larger group of researchers.

**Justification For Why Not Lower Score:**

The well-presented and clear/neat idea with extensive theoretical and empirical studies, make the work worth a spotlight.

**Metareview: Summary, Strengths And Weaknesses:**

The authors present Pseudo-supervised Contrast (PsCo), an unsupervised meta-learning framework that solves a few-shot learning problem from the constructed few-shot pseudo-tasks. The key idea of the approach is to improve pseudo-labelling progressively during meta-learning by using current and previous mini-batches which is achieved using momentum network and momentum queue.

During the discussion phase, the additional ablation and comparisons against stronger transfer learning baselines, as well as the clarifications to the paper, strengthen the submission significantly.

I agree with the consensus from the reviewers that this work could be published at ICLR.

**Note From Pc:**

if the above contains the word "oral" or "spotlight" please see: "oral" presentation means -> notable-top-5% and "spotlight" means -> notable-top-25%. As stated in our emails, we are disassociating presentation type from AC recommendations

**Summary Of Ac-Reviewer Meeting:**

N/A